# Nucleus reuniens of the thalamus contains head direction cells

**Maciej M Jankowski[1†], Md Nurul Islam[1†], Nicholas F Wright[2], Seralynne D Vann[2], Jonathan T Erichsen[3], John P Aggleton[2], Shane M O'Mara[1]***

[1]Institute of Neuroscience, Trinity College Dublin, Dublin, Ireland; [2]School of Psychology, Cardiff University, Cardiff, United Kingdom; [3]Department of Optometry and Visual Science, Cardiff University, Cardiff, United Kingdom

**Abstract** Discrete populations of brain cells signal heading direction, rather like a compass. These 'head direction' cells are largely confined to a closely-connected network of sites. We describe, for the first time, a population of head direction cells in nucleus reuniens of the thalamus in the freely-moving rat. This novel subcortical head direction signal potentially modulates the hippocampal CA fields directly and, thus, informs spatial processing and memory.

*For correspondence: smomara@tcd.ie

†These authors contributed equally to this work

Competing interests: The authors declare that no competing interests exist.

## Introduction

Nucleus reuniens (NRe), one of the largest midline thalamic nuclei, receives extensive limbic inputs and provides a bridge linking the hippocampus (especially area CA1) with medial prefrontal cortex (*McKenna and Vertes, 2004*; *Vertes, 2006*; *Prasad and Chudasama, 2013*). Its functions are not well-understood, but it has been suggested that, via these connections, NRe influences memory consolidation for spatial learning and generalisation of fear conditioning (*Eleore et al., 2011*; *Loureiro et al., 2012*; *Xu and Sudhof, 2013*). To elucidate its functions open-field single-unit recordings in NRe of freely-moving rats were made (*Mink et al., 1983*). We have found an unexpected population of NRe cells signalling head direction (HD) in the horizontal plane, independent of location within the test arena. These cells resemble HD cells in the anterodorsal and anteroventral thalamic nuclei (*Shinder and Taube, 2011*; *Tsanov et al., 2011*), the lateral mammillary nucleus (*Taube, 2007*), and certain parahippocampal regions (*Cassel et al., 2013*). NRe cells maintain head directionality during light–dark transitions, and in environments of different shape. These cells establish directionality rapidly upon first entering an environment. NRe has not, to date, been part of the traditional HD circuit, which largely originates in the dorsal tegmental nucleus of Gudden and the lateral mammillary nucleus (*Taube, 2007*; *Cassel et al., 2013*). Subsequent processing is via the anterodorsal thalamic nucleus, dorsal presubiculum, entorhinal cortex and to the hippocampus (*Su and Bentivoglio, 1990*; *Vertes et al., 2007*). Our findings, therefore, reveal a novel head direction signal potentially modulating the hippocampal CA fields and, therefore, hippocampal spatial processing (*Brandon et al., 2013*).

## Results

### Changing visual conditions from light to dark to light does not affect HD cells

Lighting conditions were systematically varied across foraging sessions for 10 cells. The animal foraged during light-dark-light sessions (each 20 min). Light removal did not affect NRe head directional activity (*Figure 1*).

**eLife digest** Whether it is foraging for food or finding its way back to its nest, an animal often needs to know which direction it is heading in. Some neurons in a mammal's brain have been shown to act like a compass, and send out nerve impulses whenever the animal points its head in a certain direction. For example, some of these neurons will fire when the animal faces northeast, but not when it faces northwest, and vice versa.

Importantly these neurons, called 'head direction' cells, do not actually measure the Earth's magnetic field. Rather, they respond to information about landmarks in the environment and the animal's movements of its head or body to work out which way the animal is facing.

Head direction cells are largely found in a closely-connected network of a few sites in the brain. However, Jankowski et al. have now discovered more of these cells in another region found deep within the centre of the brain. Measuring the nerve impulses from these neurons in rats that were moving freely around a test arena revealed that the neurons fired in exactly the same way as some other head direction cells in other regions of the brain. For example, they fired whenever the rat faced one direction, but stopped firing when it turned its head to face another.

Jankowski et al. showed that the head direction cells in this region of the brain continued to work when the lights were turned off in the test arena, or when the shape of the arena was changed from a circle to a square. These neurons began sending information about head direction as soon as the rat entered the test arena, and many continued to fire when the rat faced the same direction even when they were retested on several different days.

The head direction cells discovered by Jankowski et al. are connected to another region of the brain that is involved in remembering different locations in the environment and navigating between them. This suggests that these neurons might provide some of the information required to carry out these tasks. It also means that areas of the brain close to those that receive input from the outside world may perform more complex cognitive functions than previously thought.

## Arena shape does not affect HD cells

We transformed arena shape (circle-square-circle; *Figure 1D*; circle-square; *Figure 2A,D*). There was no effect of environmental shape changes on any HD cells (n = 10).

Transitions between light and dark, arena shape, mean head direction, mean head direction for clockwise and counter-clockwise movement, peak head direction and peak firing rate were compared using t-test for paired two samples for means with Bonferroni correction. No significant differences were observed between conditions.

## HD cells do not remap across days

22 cells were recorded across at least 2 consecutive days (15 were recorded for three or more days, and three were stably recorded for 14 or more days. Initial head direction remained stable across days, even for extended recordings), indicating no effect of time or sleep/wake cycle on their preferred directionality.

## Temporal development of HD cells

*Figure 2C,F* depict the temporal evolution of head directional firing for cumulative samples and independent time-binned samples, demonstrating head directional activity is present from the first minute of exposure to the environment (irrespective of environment shape).

## Separation angle (clockwise vs counter-clockwise) is present

Separation angle (the offset in mean peak firing rate for clockwise vs counter-clockwise head movements) is present in about 50% of recorded HD cells. There were no significant differences between mean head direction measured in degrees for clockwise and counter-clockwise movement in the *t* test for paired two samples for means in the whole population of HD cells recorded in NRe.

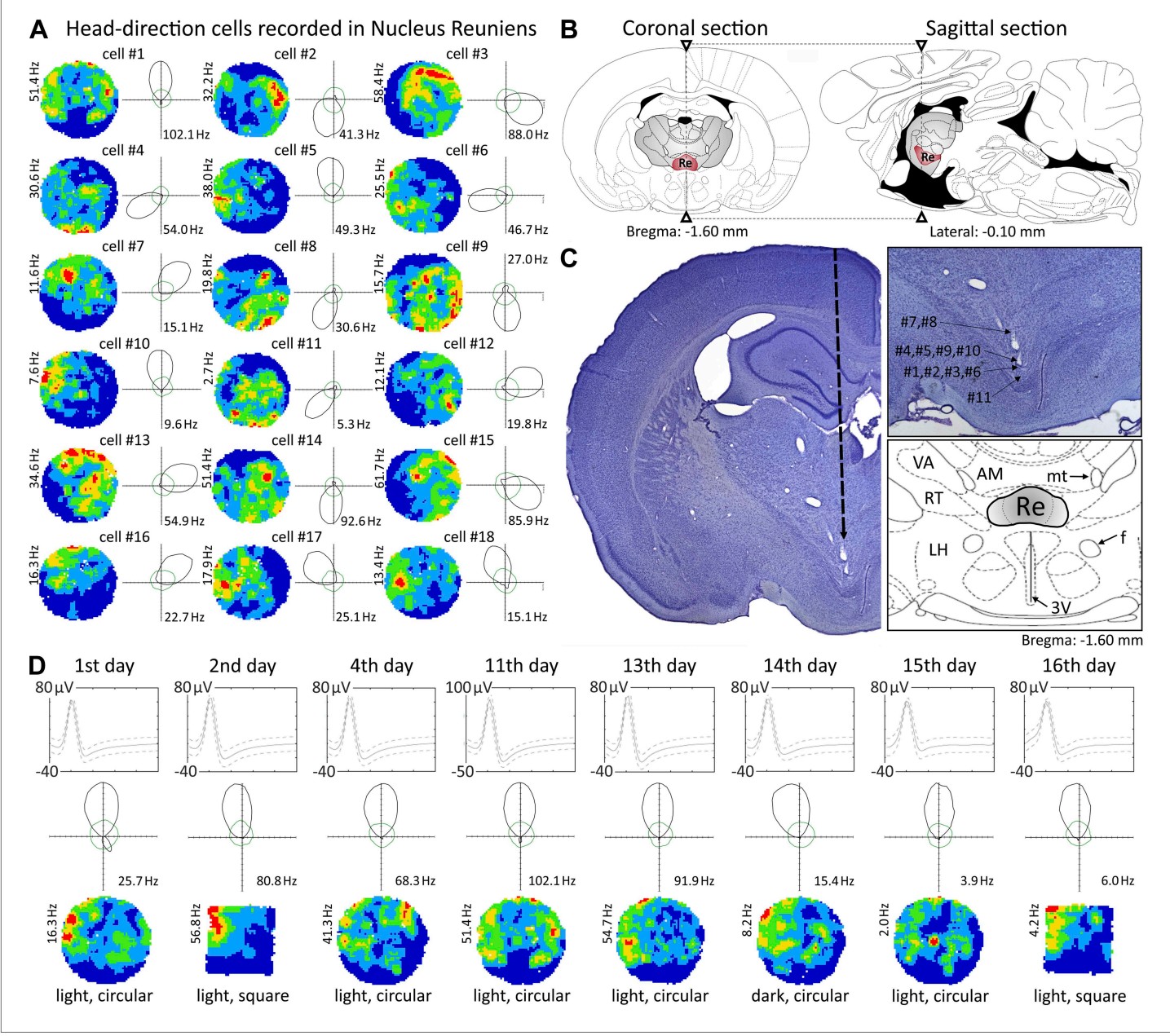

**Figure 1**. Head direction cells recorded in the nucleus reuniens. (**A**) 18 representative head direction (HD) cells in nucleus reuniens (NRe); (**B**) NRe location on a coronal (left) and corresponding sagittal (right) rat brain section (adapted from **Paxinos and Watson, 2005**); (**C**) representative histological specimen showing electrode track (left); recording positions corresponding to cell locations presented in panel a (upper right inset) showing location of NRe and detailed atlas (lower right inset); (**D**) representative recordings showing multi-day stability of HD cells: a representative cell recorded on each day of 16 days (multiple transitions from light-dark-light, and environmental transformations from circle to square to circle). The solid line is the mean spike waveform and dashed lines are M ± SD of the spike waveform. The green outline shows predicted firing rates given the proportion of time the animal spent looking in each direction, calculated according to the distributive hypothesis.

## Theta-cycle skipping cells do not carry a head-directional signal

Brandon et al. (**Paxinos and Watson, 2005**) recorded in medial entorhinal cortex (mEC) and found units firing in a fixed synchronous or anti-synchronous relationship with alternate theta cycles. We find a similar population of cells in NRe (see online material (OM) for detailed calculations and **Figure 3A,B**). These cells, which do not carry a HD signal, are found only in NRe, providing a physiological marker for electrode depth. The theta skipping index is always positive ([mean ± SD] 0.334 ± 0.127; jump

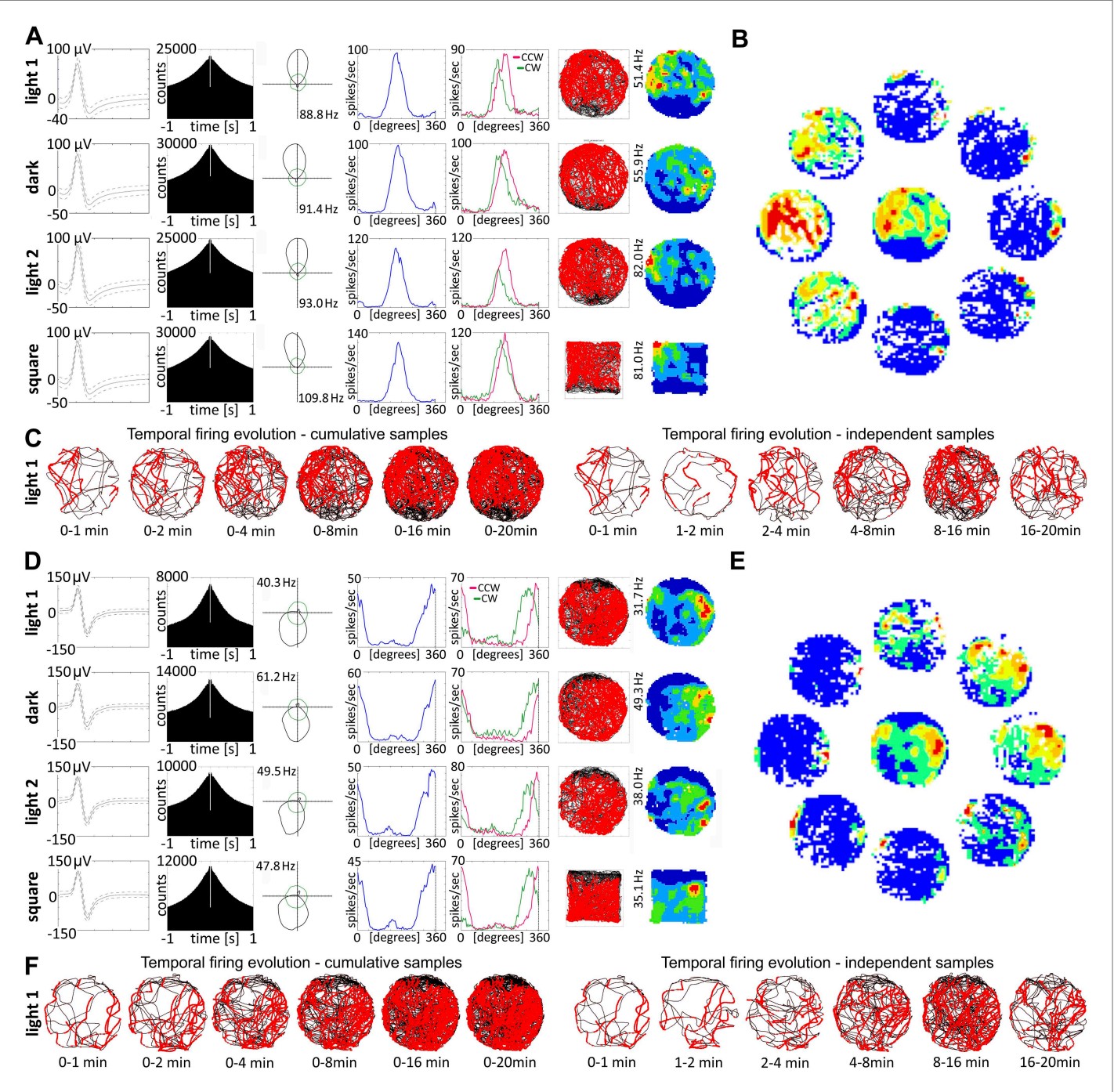

**Figure 2**. Nucleus reuniens head direction cells do not appear to be spatially modulated and are present from first exposure to the environment. (**A** and **D**) Representative waveforms for two units showing, (top to bottom), light-dark-light and circle–square transitions; from left to right, autocorrelation histogram, polar plots, tuning curves for clockwise (CW) vs counter-clockwise (CCW) head movements; (**B**) spatial analysis for cardinal orientations showing no effect of spatial position on unit activity; (**C**) temporal evolution of HD firing for cumulative samples (time ranges: 0–1, 0–2, 0–4, 0–8, 0–16 and 0–20 min) and independent time-binned samples (0–1, 1–2, 2–4, 4–8, 8–16 and 16–20 min) demonstrating that HD activity is present in the first minute of exposure to the arena; (**D**, **E**, **F**) as (**A**, **B**, **C**). Red lines are formed by continuous firing activity when the rat walks with its head directed in the preferred HD. The firing map represents −22.5° to +22.5°; as the ring was formed by eight plots each representing a 45° extent of head direction.

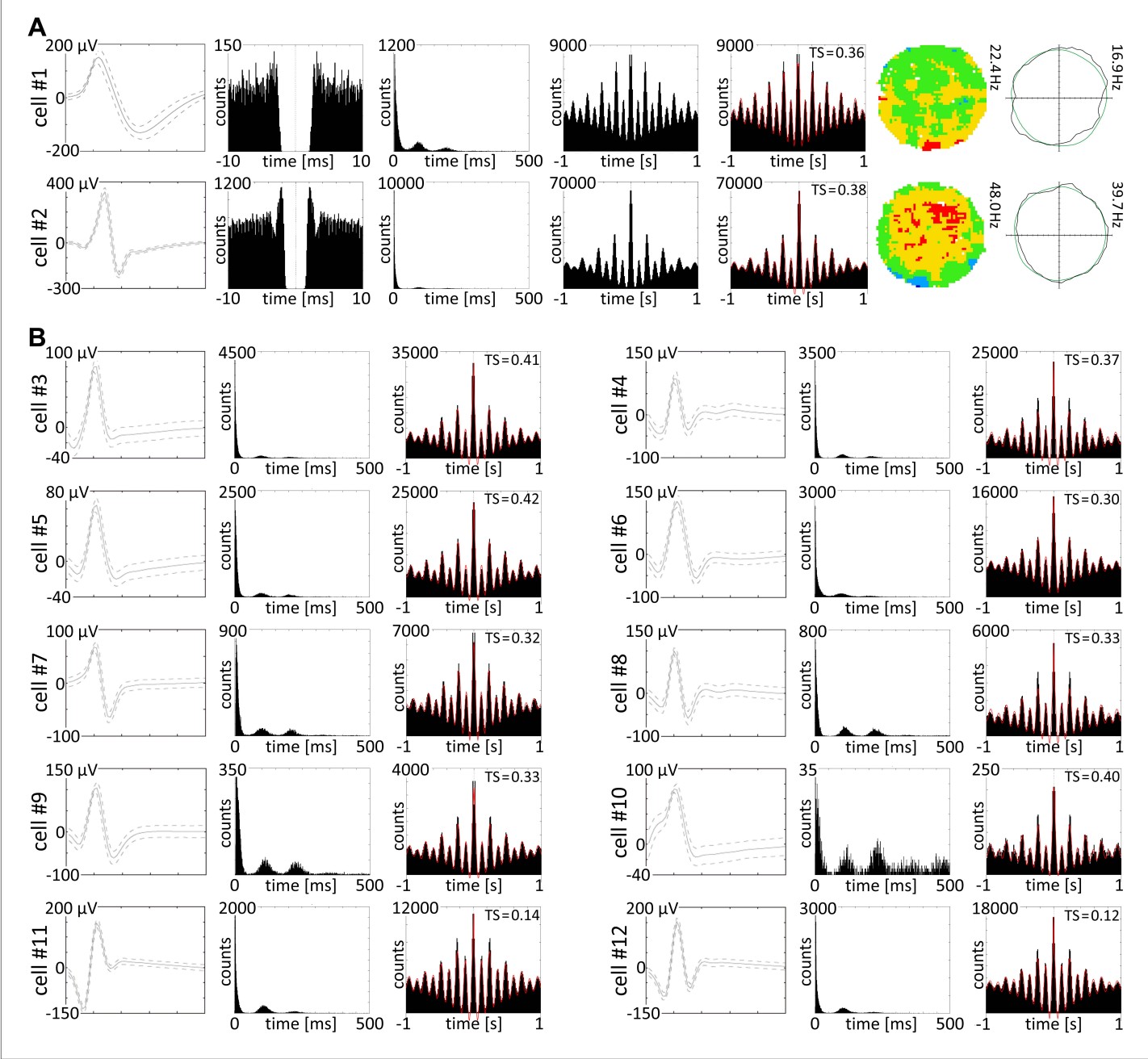

**Figure 3**. Theta skipping cells are present in nucleus reuniens. (**A** and **B**) 12 representative theta cycle skipping cells which do not carry HD signal recorded in NRe; (**A**) from the left waveform, autocorrelation per 10 ms, interspike interval histogram (ISIH), autocorrelations per 1000 ms with and without fitting to the model, firing intensity map and polar plot are presented for two representative cells. Red line: envelope of the autocorrelation histogram obtained by fitting histogram data to the equation (see OM); (**B**) waveforms, ISIHs, and autocorrelation histograms with redline envelope of data fitting to the equation (OM) for 10 theta cycle skipping cells recorded in NRe.

factor: 0.367 ± 0.075; frequency ratio: of 2.00 ± 0.018). Frequencies (mean ± SD) of the higher and lower oscillations were 8.670 ± 0.259 Hz and 4.334 ± 0.112 Hz, respectively.

## Spatial analyses

All described HD cells were analysed to test for spatial/place cell modulation (OM). In no case was any spatial modulation of the HD cell signal detected (see, e.g., *Figure 2B,E*).

## Discussion

Nucleus Reuniens has been investigated anatomically, but remains largely unexplored electrophysiologically (*Cassel et al., 2013*). The HD cells in NRe share many characteristics of other HD cells, such as those in the anterodorsal and anteroventral thalamic nuclei, the dorsal presubiculum and retrosplenial cortex (*Taube, 2007*). Current theoretical models of HD information suggest that the HD signal derives, in part, from the dorsal tegmental nucleus of Gudden and the lateral mammillary nuclei (*Taube, 2007*; *Cassel et al., 2013*). In addition to projecting to the anterodorsal thalamic nucleus, the lateral mammillary nucleus innervates NRe (*McKenna and Vertes, 2004*). NRe also receives inputs from presubiculum (postsubiculum) and retrosplenial cortex, other sites containing HD cells. This pattern of connectivity suggests various potential sources for the heading information found in NRe. The presence of spatial cells in NRe is highly significant, as it is the major source of direct thalamic projections to the CA fields of the rat hippocampus (*Su and Bentivoglio, 1990*). NRe also receives widespread cortical and limbic inputs, involving both interoceptive and exteroceptive information. This nucleus has a notable role linking medial prefrontal cortex with the hippocampus (*Prasad and Chudasama, 2013*). These NRe HD cells are, therefore, pivotally positioned to influence hippocampal spatial processing directly because of its dense, direct connections with both the prefrontal cortex and hippocampus (*Vertes, 2006*; *Prasad and Chudasama, 2013*). Finally, the theta-skipping cells may provide a pace-maker like function for synchronising some early components of the HD system.

## Materials and methods

### Online data archive

The data from which this paper was generated are archived at doi: 10.5061/dryad.j68v0 and may be accessed via http://dx.doi.org/10.5061/dryad.j68v0.

### Rats

Eleven (4–6 months) male Lister-Hooded rats (B&K, UK) weighing 420–530 g were used. Upon arrival, animals were housed individually and handled by the experimenter daily for a week before being trained in the pellet-chasing task (see below). Rats were food-restricted to 85% of their *ad libitum* body weight and kept in a temperature-controlled laminar airflow unit and maintained on a 12-hr light/dark cycle (lights on from 08:00 to 20:00 hr). Experiments were carried out in strict accordance with regulations laid out by LAST Ireland and were compliant with the European Union directives on animal experimentation (86/609/EEC).

### Behavioural testing

Experiments were conducted in a circular arena (diameter 96 cms) and square arena (60 × 60 cm). The insides of the arenas were a uniform matt black, and low-level lighting was used during light testing; all lights were extinguished during dark testing. All experiments were conducted during the day between 0900 and 2000 hr. Session lengths were typically 20 min duration. Rats performed a pellet-chasing task during the course of the experiments. During testing, 20 mg food pellets (TestDiet, 5TUL formula) were thrown in the arena at random locations ca. every 20 s. During the weeks of recordings, animals were allowed 20 g of food daily. The environment is partially curtained with a visual cue card in a constant location. We leave the rat in the environment during the LDL transitions.

### In vivo electrophysiology and surgery

Detailed descriptions of the surgical protocol and recording techniques can be found elsewhere (*Brotons-Mas et al., 2010*; *Tsanov et al., 2011*). Briefly, rats were implanted with tetrodes of either four or eight bundles of ø 25 μm platinum–iridium wires (California Fine Wire Ltd., USA) mounted onto small driveable microdrives (Axona Ltd., UK) at the following coordinates targeted at the nucleus reuniens (see *Figure 1* for histological verification): −1.60 mm posterior to bregma, −1.20 mm lateral to the midline and at angle of about 5.5°. Depth varied depending on the target structure and ranged from 4.8 to 5.6 mm below the brain surface (*Figure 1*; *Paxinos and Watson, 2005*). Rats were allowed at least 1 week of recovery post-surgery. Tetrodes were lowered slowly through the brain (maximal rates 25–50 μm/day), typically over a period of weeks to prevent tissue damage and to ensure successful NRe electrode targeting and penetration. Based on the daily record of the electrode position and post-mortem histological verification, each recording could be located along the tetrode trace.

The recording sessions took place in arenas located in the centre of the test room, which contained multiple, large visual cues made salient to allow the animals to orient themselves in the environment. An example of a NRe HD cell is provided in *Video 1*.

## Recording and statistical analysis

Standard statistical testing used Matlab scripts and Axona software. Unit identification involved several criteria. First, neurons had to be active in all conditions and had to present the same waveform characteristics (amplitude, height, and duration) in those conditions. Furthermore, units had to demonstrate a clean refractory period (>2 ms) in the inter-spike interval (ISI) histogram. Units were sorted using conventional cluster-cutting techniques and classified by the environmental manipulation to which they were exposed. Once well-defined neuronal signals were isolated and the rats explored the arena sufficiently (rats had to explore at least 90% of the open field in either session to be included in analysis to allow reliable calculation of spatial characteristics), recording commenced.

In total, 758 well-isolated units were recorded in 3 rats in NRe and the adjacent anteromedial thalamic nucleus (AM). After post-mortem histological verification, 483 cells were assigned to NRe. To select animals for analysis we set following criteria: histologically-verified electrodes should be placed in the lateral part of nucleus reuniens, therefore bypassing laterally the Rhomboid nucleus, avoiding the possibility that by mistake some cells would be wrongly assigned to NRe or the Rhomboid nucleus. Further, an observed electrophysiological criterion required co-localised HD cells and theta skipping cells in NRe. We did recordings in many animals in surrounding nuclei but as described in the present paper theta skipping cells are characteristic only for NRe, are electrophysiologically colocalised with HD cells therefore can serve as an electrophysiological marker of lateral part of NRe. Electrode tracks were localised predominantly in the lateral portion of NRe. The numbers and percentages of cells recorded in NRe were: 42 HD cells (8.7%); 19 theta cycle skipping cells (3.9%; these cells characteristically appeared in NRe, electrophysiologically co-localised with HD cells); 55 other theta modulated cells (11.3%); 23 fast firing cells (4.7%); 21 weakly-theta modulated cells (4.3%); 13 other spatially-tuned cells (2.2%). 309 cells (63.9%) were classified as unidentified low firing units–cells that did not exhibit any particular temporal or spatial properties or formed groups smaller than four cells with similar phenotype. Among unidentified low firing units 53 cells fired with maximum frequency lower than 1 Hz.

## Head direction analyses

Directional analyses were performed for all recorded cells in nucleus reuniens, 42 units in total). The rat's HD was calculated for each tracker sample from the projection of the relative position of the LEDs onto the horizontal plane. The directional tuning function for each cell was obtained by plotting the firing rate as a function of the rat's directional heading, divided into bins of 5°. The firing rate was computed based on the total number of spikes divided by the total time in that bin (*Taube et al., 1990*). To restrict the influence of inhomogeneous sampling on directional tuning, we accepted data only if all directional bins were sampled by the rat. The directionality of the HD units in the horizontal plane (measured in degrees) was normalized for comparison of the HD firing rate properties. The peak firing rate of cells that respond to different direction of heading was aligned to a HD of 180° (*Bassett et al., 2005*). The firing rate was normalized (with values between 0 and 1) with respect to the peak firing rate for each unit (*Bassett et al., 2005*). The firing rate is calculated by dividing the number of spikes by the number of visits at a particular head direction bin. The CW/CCW separation was calculated by considering a particular angular head velocity threshold (120°/s). When the rat moves at +120°/s, it was

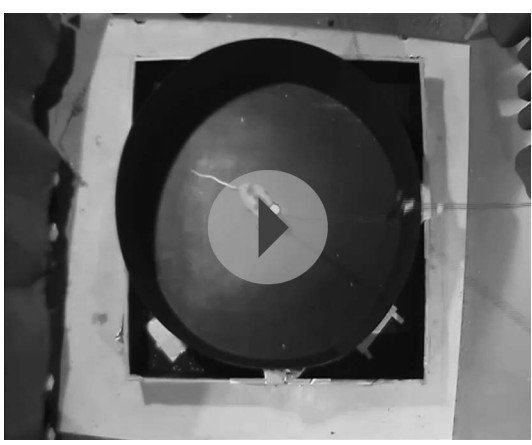

**Video 1**. An example of a well-discriminated head direction unit recorded in NRe while the rat is engaged in pellet chasing in a circular arena. The unit corresponds to unit 4 of *Figure 1A*.

placed into the CW head direction category (similarly, for CCW, −120°/s). If it coincidentally happens that spikes occur at a lower rate during slower head movements, the visit count will increase in the non-separating case, the spike count will be relatively decreased (causing a decrease in the firing rate compared to the CW/CCW separation).

## Spatial analyses

Additional analyses examined for spatial modulation of these cells. The spatial specificity (or spatial information content) is expressed in bits per spike (*Skaggs et al., 1996*). Place field size is computed as the region of the arena in which the firing rate of the place cell is above 20% of the maximum firing frequency (*Hollup et al., 2001*). A place field was identified if nine neighbouring pixels (sharing a side) were above 20% of the peak firing rate. Place field size was represented in number of pixels. The spatial selectivity of a firing field (ratio of maximal signal to noise) was calculated by dividing the firing rate of the cell in the bin with the maximum average rate by its mean firing rate over the entire apparatus (*Skaggs et al., 1996*). Average frequency is the total number of spikes divided by the total recording time and is expressed in Hz. Where recordings were conducted on successive days, units were matched based on their spike amplitude, height, and spike duration with the respective units from the control recording. Exploration was assessed by comparing the occupancy of bins and the number of visits per bin between the two recording conditions.

### Autocorrelation histograms

Autocorrelation histograms of the ISI distribution were obtained for ISI lags of −1000 ms to +1000 ms between spikes with 1 ms-wide bins. The non-normalized autocorrelation signal was fitted to *Equation 1* (an extended form of the one used by *Brandon et al. (2013)*).

$$y(x) = [a_1 \cos(\omega_1 x) + a_2 \cos(\omega_2 x)] \times \exp\left(-\frac{|x|}{\tau_1}\right) + b + c_1 \exp\left(-\frac{|x|}{\tau_2}\right) - c_2 \exp\left(-\frac{|x|}{\tau_3}\right)$$

Here, $y$ is the autocorrelation signal, $x$ is the lag, $a_1$ and $a_2$ are the amplitude of the oscillating terms with frequency $\omega_1$ and $\omega_2$ modulated by exponential with decay constant $\tau_1$. The terms $\tau_2$ and $\tau_3$ are the decay constants for the exponentials with amplitude $c_1$ and $c_2$. The parameters $a_1$, $a_2$, $b$, $c_1$ and $c_2$ were allowed to vary in the range [0, N], where N is the peak of the autocorrelation signal. The range of decay constants are, $\tau_1 = [0, 5000]$, $\tau_2 = [0, 100]$, $\tau_3 = [0, 10]$ in millisecond unit. $\omega_1$ and $\omega_2$ were varied in the range [12π, 24π] and [6π, 12π], respectively.

Assuming spikes were generated following a Poisson process, the ISI was considered to follow an exponential distribution, which also gives an exponential distribution in its autocorrelation, represented in *Equation 1* by the positive exponential component comprised of $(c_1, \tau_2)$. The initial dip in the autocorrelation for the refractory period in the ISI was given by the fast-decaying negative exponential $(c_2, \tau_3)$. The alternating low and high peaks are modelled as the superposition of two oscillations, given by two slow decaying cosine functions, as if the high peaks are generated when the oscillations are in the same-phase and the low peaks are generated when they are in anti-phase. The baseline shifts for the cosine terms and all the constant errors are interpreted in constant $b$. *Equation 1* does not hold the prior assumption that one of the periodic functions oscillates exactly at half-frequency of the other, as others have assumed (*Brandon et al., 2013*). The curve fitting followed the measurement of 'jump factor', defined as the relative contribution of the low frequency components in the higher peaks given by $a_2/(a_1 + a_2)$. Frequency ratios of the cosine functions were also measured to verify the superposition model of the *Equation 1*. Theta cycle skipping index, TS was measured by Equation 4; *Brandon et al., 2013*.

As described (*Brandon et al., 2013*), TS range should be [−1 to +1]. A positive TS is present if the second peak after the centre peak is larger than the first one; a negative TS is not expected according to the model in *Equation 1* if it is fitted for the theta-skipping cycle cell with an alternative low and high peaks and low peaks appearing first after the centre peak. We did not find any negative TSs (as can be seen in the raw autocorrelograms; see supplementary figures). The greater the TS, the greater the larger peak is jumped from the theta-modulated signal, caused by the interference of the second oscillation. We measured this effect with an alternative index 'jump factor' defined as $a_2/(a_1 + a_2)$, which provides a direct measurement of the relative contribution of the two oscillations. A jump factor >0.5 means the contribution of the slow varying interfering oscillation in the jump is greater than the

theta-range oscillation. If the frequency ratio follows a 2:1 ratio (or closer), then the mathematical basis of using two cosine functions of different frequency ranges to obtain consecutive in-phase or anti-phase superposition to yield lower and higher peaks respectively, is verified.

## Histological analyses

On completion of the recording studies, the rats received an overdose of anaesthetic (1.5 g of ure-thane (Sigma-Aldrich, Dublin, Ireland) dissolved in 4.5 ml water) and were then perfused intracardially with 250 ml of 0.1 M phosphate-buffered saline (PBS) at room temperature followed by 350 ml of 4% paraformaldehyde in 0.1 M PBS at ~4°C, after which the brains were removed and placed in 4% para-formaldehyde (for at least 72 hr). Brains were blocked, placed on a freezing platform, and 40 µm cor-onal sections were cut with a sledge microtome (Leica 1400). Two, alternate series that used all sections were taken through the rostral thalamus. One series was mounted directly onto gelatine-subbed slides, and then allowed to dry overnight before staining with cresyl violet, a Nissl stain. The second series was immunologically reacted with the neuronal marker α-NeuN (MAB 377; Chemicon, Watford, UK), then with a secondary horse anti-mouse rat adsorbed antibody (AI-2001; Vector Laboratories Ltd, Peterborough, UK) and subsequently visualised with Vector Elite ABC (PK-6100; Vector Laboratories Ltd) and diaminobenzidine. A Leica DM5000B microscope with Leica DFC310FX digital camera and Leica Application Suite image acquisition software was used for brightfield microscopy. NeuN pro-vides a stain assumed to be selective for neurons, that is it does not label glial cells. As a consequence it can sometimes make it easier to see underlying cytoarchtectonic features.

Recording positions were determined by calculating distance above the deepest electrode posi-tion, and calculating distance below the first penetration into the tissue. The electrodes often caused tissue distortion and this was carefully allowed for in the position calculations. Positions of recorded cells were estimated as follows: theoretical positions of electrodes tips and NRe borderlines were estimated by reference to atlas (*Paxinos and Watson, 2005*) and reconstructed histological speci-mens. The electrodes sometimes caused tissue distortion even with very slow penetrations, and this was allowed for in the position calculations. The position of electrodes below brain surface was known for each recording session and expressed in µm, thereby allowing estimates of each cell position to be derived.

## Acknowledgements

This work was supported by The Wellcome Trust.

## Additional information

### Funding

| Funder | Grant reference number | Author |
|---|---|---|
| Wellcome Trust | WT092480 | Shane M O'Mara |

The funder had no role in study design, data collection and interpretation, or the decision to submit the work for publication.

### Author contributions

MMJ, Acquisition of data, Analysis and interpretation of data, Drafting or revising the article; MNI, SDV, JTE, Analysis and interpretation of data, Drafting or revising the article; NFW, Analysis and interpretation of data; JPA, SMO'M, Conception and design, Analysis and interpretation of data, Drafting and revising the article

### Ethics

Animal experimentation: Experiments were conducted in accordance with European Community directive, 86/609/EC, and the Cruelty to Animals Act, 1876, and followed Bioresources Ethics Committee, Trinity College, Dublin, Ireland, as well as LAST Ireland and international guidelines of good practice. Surgery was conducted under ketamine/xylazine anaesthesia, an appropriate post-surgery monitoring and analgesia regime was in place, and every effort was made to minimize suffering.

# Additional files

## Major dataset

The following dataset was generated:

| Author(s) | Year | Dataset title | Dataset ID and/or URL | Database, license, and accessibility information |
|---|---|---|---|---|
| Jankowski MM, Islam MN, Wright NF, Vann SD, Erichsen JT, Aggleton JP, O'Mara SM | 2014 | Data from: Nucleus Reuniens of the Thalamus Contains Head Direction Cells | http://dx.doi.org/10.5061/dryad.j68v0 | Available at Dryad Digital Repository under a CC0 Public Domain Dedication. |

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
