## [Decision Letter]

Thank you for sending your work entitled “Nucleus Reuniens of the Thalamus Contains Head Direction Cells” for consideration at *eLife*. Your article has been favorably evaluated by Eve Marder (Senior editor) and 3 reviewers, one of whom is a member of our Board of Reviewing Editors, and one of whom, Bruno Poucet, has agreed to reveal his identity.

The Reviewing editor and the other reviewers discussed their comments before we reached this decision, and the Reviewing editor has assembled the following comments to help you prepare a revised submission.

The authors report the existence of head direction (HD) cells in nucleus reuniens (NRe) of rats exploring an arena. Their conclusion is that, because this thalamic nucleus is in a strategic position between the medial prefrontal cortex (mPFC) and the hippocampus, these HD cells might provide the hippocampus with directional information that orients the hippocampal map. This is therefore an original and important finding. The paper is concise and provides most of the information required to understand and interpret the main results. It is clearly accessible and should be of interest to researchers studying the neural circuits of spatial information processing.

Major concerns:

1) Are these cells pure HD cells or are they better described as place x HD cells, which have been reported in other brain areas as well, most notably by Cacucci et al. (2004, J Neurosci). Looking at the two cells displayed in Figure 2, both cells appear to be more place x HD cells than just HD cells. This is particularly true for the cell depicted in E. Upon further perusal of the 18 cells depicted in Figure 1, most of them can be more described as place x HD cells than pure HD cells. The authors need to examine this possibility more critically.

2) While these results are interesting, they are difficult to place in context in terms of their functional significance. Little is known about the nucleus reuniens and it is not traditionally associated with other areas where HD cells have previously been reported. Although a little information is provided about the anatomical connections and functional significance of the reuniens nucleus, but the manuscript would be improved if a little more information along these lines was provided. This will aid in putting into context the significance of the current findings. Also, it is not clear how these signals contribute to spatial or other information processing, or whether these cells could have very different properties if the animal was, for example, learning something. Similarly, there is a description of theta-skipping cells, and nothing is said about how these might contribute.

3) Figure 2: The firing rate maps in B and E appear distorted in shape (not true circles). At first I thought that this may be due to the fact that the animal would have difficulty getting into certain positions with its head oriented one way. This aspect is very evident in the original HD papers by Taube et al. (1990, J Neurosci). However, the center map should not show this distortion, but it does. Thus, this calls into question somewhat the accuracy of the different orientation maps because there really should be a zone for each map where the animal should have difficulty getting into certain HD orientations at specific locations in the arena. This area of difficulty should shift from map to map as reported in the Taube et al. paper. Why doesn't this occur in these maps?

4) The characterization of these cells is very good, but we are not provided with any information about how prevalent they are in this nucleus, and it would help to have comparisons of their prevalence in n. reunions and other brain areas.

---

## [Author Response]

*1) Are these cells pure HD cells or are they better described as place x HD cells, which have been reported in other brain areas as well, most notably by Cacucci et al. (2004, J Neurosci). Looking at the two cells displayed in*
Figure 2*, both cells appear to be more place x HD cells than just HD cells. This is particularly true for the cell depicted in E. Upon further perusal of the 18 cells depicted in*
Figure 1*, most of them can be more described as place x HD cells than pure HD cells. The authors need to examine this possibility more critically*.

This is an issue we have also spent a considerable amount of time on, in order to ensure that we have represented the data appropriately. Each individual circle in the outer ring of Figure 2 shows the firing intensity map at the corresponding head direction. If the cell is HDxPlace, then the firing intensity map would/should show focused activity at particular directions that would vary systematically, instead of showing distributed activity (Cho and Sharp 2001, Sharp 1996). As the cell fires only in a range of directions, the firing map shows the activity in the entire arena/or most of it only at that direction range. Other places remain mostly inactive, and thus we conclude that these cells are almost purely HD cells. Furthermore these cells do not form place fields in the standard analysis – as described in the Methods section. We have checked Cacucci et al., 2004, J Neurosci, http://www.jneurosci.org/content/24/38/8265.full.pdf); they used the same analysis but their results appear to differ to ours - there is constant place representation. It is possible that there is a residual place representation here, but statistically current methods do not detect it.

*2) While these results are interesting, they are difficult to place in context in terms of their functional significance. Little is known about the nucleus reuniens and it is not traditionally associated with other areas where HD cells have previously been reported. Although a little information is provided about the anatomical connections and functional significance of the reuniens nucleus, but the manuscript would be improved if a little more information along these lines was provided. This will aid in putting into context the significance of the current findings. Also, it is not clear how these signals contribute to spatial or other information processing, or whether these cells could have very different properties if the animal was, for example, learning something. Similarly, there is a description of theta-skipping cells, and nothing is said about how these might contribute*.

We now add a sentence to the Discussion to provide contextual information:

“These NRe HD cells are, therefore, pivotally positioned to influence hippocampal spatial processing directly because of its dense, direct connections with both the prefrontal cortex and hippocampus(19; 11). Finally, the theta-skipping cells may provide a pace-maker like function for synchronising some early components of the HD system.”

*3)*
Figure 2*: The firing rate maps in B and E appear distorted in shape (not true circles). At first I thought that this may be due to the fact that the animal would have difficulty getting into certain positions with its head oriented one way. This aspect is very evident in the original HD papers by Taube et al. (1990, J Neurosci). However, the center map should not show this distortion, but it does. Thus, this calls into question somewhat the accuracy of the different orientation maps because there really should be a zone for each map where the animal should have difficulty getting into certain HD orientations at specific locations in the arena. This area of difficulty should shift from map to map as reported in the Taube et al. paper*. *Why doesn't this occur in these maps?*

We thank the referees for spotting this. These maps were deformed slightly by an auto-setting in Corel during preparation of the Figure and the corrected map is now presented.

*4) The characterization of these cells is very good, but we are not provided with any information about how prevalent they are in this nucleus, and it would help to have comparisons of their prevalence in n. reunions and other brain areas*.

The information about prevalence of cells is now detailed in the ‘Recording and statistical analysis’ section. The issue of prevalence with other brain areas is a more difficult issue to deal with in the limited space we have available, but we do agree that it is an important issue (especially for theoretical model-building reasons).